# A Novel Artificial Coronary Plaque to Model Coronary Heart Disease

**DOI:** 10.3390/biomimetics9040197

**Published:** 2024-03-26

**Authors:** Philipp Lindenhahn, Jannik Richter, Iliyana Pepelanova, Bettina Seeger, Holger A. Volk, Rabea Hinkel, Bernhard Hiebl, Thomas Scheper, Jan B. Hinrichs, Lena S. Becker, Axel Haverich, Tim Kaufeld

**Affiliations:** 1Department of Cardiothoracic, Transplantation and Vascular Surgery, Hannover Medical School, 30559 Hannover, Germany; haverich.axel@mh-hannover.de (A.H.); kaufeld.tim@mh-hannover.de (T.K.); 2Department of Small Animal Medicine and Surgery, University of Veterinary Medicine Hanover, 30559 Hannover, Germany; 3Institute of Technical Chemistry, Leibniz University of Hannover, 30167 Hannover, Germany; richter@iftc.uni-hannover.de (J.R.); scheper@iftc.uni-hannover.de (T.S.); 4Institute for Food Quality and Food Safety, University of Veterinary Medicine, 30559 Hannover, Germany; bettina.seeger@tiho-hannover.de; 5Department of Laboratory Animal Science, Leibnitz-Institut für Primatenforschung, Deutsches Primatenzentrum GmbH, Kellnerweg 4, 37077 Göttingen, Germany; 6DZHK (German Centre for Cardiovascular Research), Partner Site Göttingen, 37075 Göttingen, Germany; 7Institute for Animal Hygiene, Animal Welfare and Farm Animal Behavior, University of Veterinary Medicine, 30559 Hannover, Germany; bernhard.hiebl@tiho-hannover.de; 8Department of Diagnostic and Interventional Radiology, Hannover Medical School, Carl-Neuberg-Str. 1, 30625 Hannover, Germany; hinrichs.jan@mh-hannover.de (J.B.H.); becker.lena@mh-hannover.de (L.S.B.)

**Keywords:** plaque, atherosclerosis, coronary heart disease

## Abstract

Background: Experimental coronary artery interventions are currently being performed on non-diseased blood vessels in healthy animals. To provide a more realistic pathoanatomical scenario for investigations on novel interventional and surgical therapies, we aimed to fabricate a stenotic lesion, mimicking the morphology and structure of a human atherosclerotic plaque. Methods: In an interdisciplinary setting, we engineered a casting mold to create an atherosclerotic plaque with the dimensions to fit in a porcine coronary artery. Oscillatory rheology experiments took place along with long-term stability tests assessed by microscopic examination and weight monitoring. For the implantability in future in vivo setups, we performed a cytotoxicity assessment, inserted the plaque in resected pig hearts, and performed diagnostic imaging to visualize the plaque in its final position. Results: The most promising composition consists of gelatin, cholesterol, phospholipids, hydroxyapatite, and fine-grained calcium carbonate. It can be inserted in the coronary artery of human-sized pig hearts, producing a local partial stenosis and interacting like the atherosclerotic plaque by stretching and shrinking with the vessel wall and surrounding tissue. Conclusion: This artificial atherosclerotic plaque model works as a simulating tool for future medical testing and could be crucial for further specified research on coronary artery disease and is going to help to provide information about the optimal interventional and surgical care of the disease.

## 1. Introduction

Atherosclerosis is a chronic inflammatory disease of the blood vessels with a characteristic narrowing of the affected blood vessel, potentially resulting in complete occlusion culminating in a complete obstruction of the arterial lumen [1,2,3]. Stenosis of the coronary arteries leads to a lack of blood supply to the heart and is called coronary heart disease (CHD). CHD is the most common type of heart disease, and atherosclerosis remains the most important etiology of cardiovascular morbidity and mortality among symptomatic patients [4]. The occurring symptoms of the disease are very diverse and depend on the localization and the degree of change in the vessel. In consequence, atherosclerosis and the acute events (myocardial infarction and stroke) are responsible for over 50% of deaths occurring in Western countries [5,6].

Researchers are therefore increasingly focusing on the development of new artificial plaque models and the testing of these artificial plaques in terms of functionality and reproducibility as they are becoming more and more important [7].

### 1.1. Plaque Components

The pathogenesis of plaque development leads to an intense debate among researchers. As such, the role of inflammation and lipid deposition are discussed, as well as theories whether the process is initiated via intimal damage or by changes in the outer, adventitial layers of the respective artery. Irrespective of the initial hit on the vessel wall, proliferation of vascular smooth muscle cells and calcifications are common trails in plaque formation.

The knowledge of the exact composition of the human atherosclerotic plaques and their respective behavior are crucial for atherosclerosis research because damage to the vessels and the supplying organs can be reduced or completely prevented [8]. Basically, atherosclerotic plaques consist of an accumulation of cells, lipids, calcium, collagen, and inflammatory infiltrates [9].

The composition and mechanical properties of plaques vary considerably as atherosclerosis proceeds; these properties can be determined using imaging techniques [10,11], and plaques react differently to pressure, depending on how they are composed [12]. A general differentiation in composition must be made according to the plaque developmental stage [13]. Initial lesions, such as the “fatty streak” lesions, are lipid-rich and therefore rather soft. In contrast, enlarged lesions have a higher content of fibrous material compared to early plaques due to the recruitment of smooth muscle cells (SMCs) [7]. As time progresses, this proportion of SMCs increases, and in very late plaques, more calcium is embedded. This process makes these plaques harder and is often referred to as “hardening of the arteries” [14]. International studies show differences in the exact composition of atherosclerotic plaque material due to different plaque stages [1,15].

The lipid content largely consists of cholesterol esters. Depending on the phase and location of the atherosclerotic plaques, the most common esters are cholesterol palmitate with up to 40%, cholesterol oleate with up to 41%, and cholesterol linoleate with up to 38% within the lipid-rich portion. This finding is not surprising, because these three are the most common natural cholesterol esters [16].

The detailed examination of the calcium content led to the result that it is calcium phosphate, more precisely carbonate-containing hydroxyapatite with the empirical formula Ca_10_(PO_4_,CO_3_)6(OH)_2_. In the case of calcification of atherosclerotic plaques, it was observed that carbonate-containing hydroxyapatite can occur both as microcalcification and in the form of larger calcification particles. The microcalcifications seem to serve as a precursor to the large, calcified areas [17].

### 1.2. State of the Art—Artificial Plaque Compositions

Most of the earlier artificial atherosclerotic plaque (AAP) designs have been composed of inorganic materials, which do not match the organic properties of soft tissues in arteries. Other colleagues used materials such as glass [18,19], silicone-based elastomeric materials [20], acrylic or elastomeric materials, or hydrogel moldings [21]. One big disadvantage was the mono-layering of the models that does not reproduce the complex wall structure of plaques. Other attempts have also been made to directly print vascular phantoms by 3D Polyjet printers, using either hard (acrylic) or elastomeric materials (polyurethane) [21] or by hydrogel molding, using polyvinyl alcohol (PVA) [22]. Common is also the use of butter and a silicone sealant with incorporated animal bone fragments [23] or the use of plaster materials and hydroxyapatite blocks [24]. Alginate and agarose gels as well as collagen are similarly popular to produce artificial platforms or matrices [25,26].

The pathological modifications related to the development of atherosclerotic plaques within arterial vessels result in significant alterations to the mechanical properties of the diseased arterial wall. There are several methods available to characterize the mechanical behavior of atherosclerotic plaque tissue, and it is the aim of this paper to review the use of uniaxial mechanical testing [7].

The objective of this study was to build an AAP by determining the mechanical properties of plaque tissue after surgical removement. By adjusting the components of the AAP, we used a gelatin matrix from porcine skin mimicking the fibrous, cellular content, mixed with the lipid component of cholesterol and phospholipids from soybean, hydroxyapatite, and finely grained calcium carbonate. We performed different tests to get closer to the human atherosclerotic plaque. This study measured the cytotoxicity of the AAP and established a washing procedure for application in later in vivo/ex vivo experiments. The AAP is intravascularly pushable like a stent to the blood vessel of interest, creating a partial occlusion of the lumen. We tested the application into the left anterior descending artery (LAD) of removed porcine hearts.

To enhance the outcomes of the interventional procedures such as angioplasty and stenting, a broad understanding of the mechanics of diseased arteries is crucial [27]. An understanding of both lumen gain and vessel injury post-stenting has led to a greater interest in modeling stent–plaque–artery interactions and surgical procedures [27,28,29,30,31,32,33,34].

## 2. Material and Methods

In the first step, suitable casting molds were made to produce the AAP in the desired shape and dimensions and in sufficient numbers. It is desirable that the molds are reusable and easy to use. The second challenge of this work was the development and testing of the correct plaque model material. For this purpose, different compositions of different gel matrices, lipids, and calcium salts in varying concentrations were created and examined. Thereafter, suitable candidates were tested for their handling with the casting mold, characterized in more detail using microscopic and rheological methods, and compared with real human arterial plaque samples regarding the mechanical properties. For the favored plaque model compositions, the long-term stability and sterilizability were then examined so that the ideal candidate was finally obtained.

### 2.1. Plaque Model Fabrication

Unless stated otherwise, all chemicals and materials were purchased from Merck KGaA, Germany. A homogenous lipid phase was prepared by mixing 10% *w*/*v* cholesterol with 10% *w*/*v* phospholipids (from soybean). This lipid component was used to formulate the model plaques. The recipe of the atherosclerotic model consists of a gelatin matrix (15% *w*/*v*, type A from porcine skin, bloom factor 300), mixed with the lipid component (2% *w*/*v*) and containing 5% *w*/*v* hydroxyapatite (HA) and 4% *w*/*v* finely grained CaCO_3_ (calcium carbonate, Roth, Germany).

The model plaque mixture was prepared by homogenous mixing of all components and rapid transfer to the cylindrical mold. The molds were consequently placed for gelation at 4 °C overnight. The levels of individual components like gelatin, lipid mix, HA, and CaCO_3_ were varied during the investigation to evaluate how they affect plaque strength. The recipe described above represents the optimized version, with mechanical properties comparable to the donor plaques.

The crosslinking procedure was performed with three different methods. In the direct crosslinking procedure (DC), 0.5% *v*/*v* glutaraldehyde (GA) was added directly to the plaque mixture and filled into the casting mold. Gelation and crosslinking occurred at 4 °C overnight. In the double direct crosslinking procedure (DDC), 0.5% *v*/*v* GA was also used together with gelation at 4 °C for crosslinking overnight. Afterward, the plaque was removed from the mold and immersed in a 1% *v*/*v* GA bath for double crosslinking (4 °C and 24 h). Finally, in the indirect crosslinking procedure (IDC), the plaque mixture was filled into the mold without GA and gelation was performed at 4 °C overnight. On the next day, the cast plaque was removed from the mold and incubated in a GA solution (1% *v*/*v*) for 24 h for complete crosslinking.

For rheology measurements, the model plaque mixture was cast into silicone molds of the dimensions of the rheometer parallel-plate geometry (20 mm diameter and 1 mm thickness) and treated in identical fashion as described above for gelation and crosslinking.

### 2.2. Plaque Casting Mold

The plaque casting mold was manufactured from Teflon by the mechanical workshop of the Institute of Technical Chemistry, Leibniz University Hannover and consisted of two hollow cylinder halves, two caps, and a metal rod. The mold served as a template for the fabrication of artificial plaques of the desired dimensions. The mold design, as well as the casting procedure, is illustrated in Figure 1.

### 2.3. Donor Plaques

The measured atherosclerotic plaques were samples obtained with endarterectomy from three different patients with informed consent. Two donor samples were from the coronary artery, and one was from the carotid artery.

### 2.4. Oscillatory Rheology

Oscillatory rheology experiments were performed with a modular rheometer MCR 302, equipped with parallel-plate geometry (Anton Paar, Austria). The artificial plaque mixture was prepared in silicone molds in the required sample geometry (20 mm diameter and 1 mm thickness). The plaque samples were then carefully placed on the lower plate of the rheometer, and the upper plate was lowered until a contact force of 1 N was established. Amplitude sweeps were performed at a constant frequency of 1 Hz and by a deformation variation between 0.1 and 100%. Sample slip was prevented using a profiled measuring plate of 20 mm diameter. All measurements were performed at least in triplicates at 37 °C.

The donor plaques were also measured with oscillatory rheology. For this purpose, the sample material was cut into the desired measuring geometry with a scalpel. The sample was pressed between the measuring plates of the rheometer with a contact force of 1 N. The shear modulus was determined by a time sweep experiment with constant deformation and frequency (0.5% amplitude and 1 Hz). Here, as well, measurements were performed at 37 °C.

### 2.5. Microscopic Imaging and Stability Measurements

The visual characterization of the artificial plaque model was performed with a digital 3D microscope VHX-5000 (Keyence, Osaka, Japan) using a magnification of 20× and 200×. Long-term stability of the artificial plaques was assessed by microscopic examination and by weight monitoring. For this purpose, freshly prepared plaques were weighed and then incubated in sterile phosphate-buffered saline (PBS) at 37 °C. Plaques were removed at regular intervals, and excess liquid was carefully blotted with a tissue, after which samples were re-weighed with an analytical balance (Sartorius, Göttingen, Germany) or examined microscopically. All experiments were performed in triplicates. The long-term stability experiment was performed for a duration of 49 days.

### 2.6. Neutralization of Excess GA

For the conduction of in vitro tests, the fabricated artificial plaques were subjected to a washing procedure, to neutralize and remove excess GA, which is known for its toxic properties [35]. The washing procedure was performed using gentle stirring with 8% *w*/*v* glycine in 0.1 M acetate buffer, pH 4.5 for 48 h at 37 °C [36]. The excess free amino acid neutralizes GA by reacting with all remaining aldehyde groups. Afterward, plaques were washed in sterile PBS, pH 7.4 for 24 h. Consequently, the neutralized plaques were used for cell assays.

### 2.7. Cytotoxicity Test: MTT Assay Protocol for Determining the Cytotoxicity of Plaques

Extracts of the plaques were used for cytotoxicity testing according to ISO 10993-5. Plaques were extracted in Dulbecco’s Modified Eagle’s Medium containing phenol red (DMEM, Merck, Darmstadt, Germany) supplemented with 10% *v*/*v* fetal bovine serum (FBS, Biochrom, Berlin, Germany) for 24 ± 2 h at 37 °C, under permanent shaking (85 rpm) in a glass vessel. The material to extraction medium ration was 0.2 g/mL. Extraction medium without addition of sample material was incubated in parallel and served as a viability control.

The Chinese hamster lung fibroblast cell line V79MZ [37] was used for the cytotoxicity assessment of the plaques. Cells were cultured under standard cell culture conditions (37 °C, 5% CO_2_, and 95% rel. humidity). Cells were maintained in DMEM, supplemented with 10% *v*/*v* FBS, 2 mM L-glutamine (Gibco, Thermo Fisher Scientific, Waltham, MA, USA), and 100 IU/mL penicillin/100 μg/mL streptomycin (Gibco, Thermo Fisher Scientific).

Cytotoxicity assessment was performed based on ISO 10993-5 in the MTT assay with modifications. In short, 8000 V79MZ cells (passage number < 25) were seeded per well (96-well-plate) and grown for 24 ± 2 h until they reached 80% confluence. Then, cells were treated with extraction medium and its serial dilutions in cell culture medium (75%, 50%, and 25% extraction medium). The medium incubated without plaques served as viability control. In addition, 0.01% *v*/*v* Triton X-100 (Roth, Karlsruhe, Germany) in cell culture medium was used as dead control. Furthermore, a cell-free control with 100% extraction medium was carried out to exclude direct reduction in 3-(4,5-Dimethylthiazol-2-yl)-2,5-diphenyltetrazolium bromide (MTT) by the sample extract.

The extraction medium did not directly reduce MTT. All treatments were performed in four technical replicates. After 24 ± 2 h incubation, cells were washed three times with phosphorate-buffered saline and medium was changed to freshly prepared assay medium (50 µL per well) DMEM without phenol red with 1 mg/mL MTT. After 2 h incubation time, 100 µL isopropanol was added. Afterward, the activity of intracellular dehydrogenases was assessed by quantification of produced formazan calorimetrically at a wavelength of 570 nm with a reference wavelength of 650 nm. All samples with viability of 70% or more were non-cytotoxic according to ISO 10993-5.

## 3. Results

### 3.1. Effect of Composition on Plaque Material Properties and Appearance

The artificial plaque model consists essentially of a bulk hydrogel containing a dispersed cholesterol phase and insoluble calcium minerals, which mimic plaque calcification. The composition of the plaque model was based on a known atherosclerotic plaque structure and was further optimized by studying how individual artificial plaque components influence final plaque mechanical properties.

Gelatin hydrogel was selected to represent the dominant component of atherosclerotic plaque, which consists mainly of collagen I fibrous tissue together with cellular and necrotic elements [1]. The lipid component of atherosclerotic plaque was modeled by cholesterol emulsified with phospholipids [16]. The calcium in atherosclerotic plaque is in the form of carbonated hydroxyapatite [38]. For this reason, we included hydroxyapatite in the recipe, with finely grained calcium carbonate added for the creation of coarser regions of uneven topology, characteristic of the progressive calcification of atherosclerotic lesions [39].

The rheological studies show that the artificial plaque exhibits elastic properties, behaving like a solid gel with G′ higher than G″ at 37 °C (Figure 2A). Gels were stable to deformations of about 10% regardless of the formulation. The gelatin concentration had a pronounced effect on plaque material properties, with concentrations above 10% *w*/*v* leading to stronger gels with a higher shear modulus. The high viscosity of the plaque recipes above 10% led to more inhomogeneous mixtures, a fact which is mirrored in the higher standard deviation between samples. Recipes with concentrations above 15% gelatin did not allow for efficient mixing, handling, and filling in the casting mold. The increase in the lipid component was also shown to correlate with stronger mechanical properties of the plaque model. This effect has been reported for protein gels and is related to the oil emulsion droplets acting like a stabilizing filler within the protein network [40,41].

The highest impact on the final plaque mechanical properties was achieved not by varying the material composition, but by optimizing the crosslinking procedure for the plaque. GA crosslinks proteins via a reaction of its aldehyde groups to free amino groups in lysine residues of polypeptide chains. Three different crosslinking procedures were investigated. In the first procedure, called direct crosslinking (DC), the GA crosslinking agent was added directly to the plaque mixture and crosslinked overnight at 4 °C. In the double crosslinking (DDC) process, a second crosslinking step was performed with the plaque after the first crosslinking, and finally, in indirect crosslinking (IDC), the plaque was manufactured solely by physical gelation at 4 °C, followed by subsequent incubation in the GA solution for crosslinking. As can be seen from Figure 2, the IDC procedure resulted in artificial plaques whose material properties are in the range of the analyzed donor plaques. The physical gelation of gelatin at 4 °C results in a compact hydrogel network, caused by a coil-to-helix transition of the polypeptide chains. This network is fixed into position by the GA crosslinking, resulting in a highly durable plaque structure by the IDC method. In contrast, during the DC and DDC procedures, the quick crosslinking GA reaction occurs already at room temperature, thus effectively crosslinking polypeptide chains which have not associated completely into a compact gel structure. The physical gelation of gelatin is a thermodynamic phase transition and is a function of both the cooling rate and the polymerization time [42]. The tougher plaques obtained by the DDC and IDC methods, in comparison to the DC method, can thus also be attributed to longer treatment times and/or the use of GA at a higher concentration. It has been reported that 1% GA solutions are sufficient for complete gelatin crosslinking [43,44]. For this reason, the use of higher GA concentrations for boosting mechanical properties of the artificial plaques even further was not pursued in this study.

The donor plaques were also analyzed by oscillatory rheology and displayed highly variable viscoelastic moduli as can be seen in Figure 2D. This high variation in the mechanical properties of plaque has been extensively reported in the literature, both between specimens and in specimens, with differences attributed to the segment of the plaque, anatomical location, histological classification, and measurement method [27,45]. It must be noted that most references in the literature report the *Young modulus* of atherosclerotic plaques obtained by radial compressive testing or uniaxial tensile testing, while the measured values here represent shear moduli. Young modulus and shear modulus are related and can be approximated to one another through the equation  E=3G [46]. This indicates that the artificial plaque created by the IDC method displays a *Young modulus* of ca. 79.9 kPa.

A microscopic investigation of the manufactured plaques revealed that the desired proportions were faithfully replicated by the casting technique. The plaques exhibited opaque coloration, with insoluble particles of HA (Figure 3) or larger calcium carbonate islands visible (Figure 4). The presence of HA did not affect plaque material properties. The inclusion of calcium carbonate into the plaque formulation was intended to create a rougher topology, with crystalline islands of mineralized calcium on the surface, which replicate the irregular structure of atherosclerotic plaques. These structures would create additional friction on vascular walls, thus more effectively simulating complications of stenting. A variation in calcium carbonate content also did not affect plaque mechanical properties in a significant way (Figure 4E).

Both HA and CaCO_3_ did not reinforce the hydrogel network and did not lead to tougher plaque structures at the concentration levels and particle sizes used in these experiments. The use of HA nanoparticles can represent an additional strategy to boost the mechanical properties of the plaque model, as this will allow the creation of a tough nanocomposite, displaying strengthening ionic interactions between the calcium ions and the charged protein residues [47]. The main contributor to the artificial plaque’s viscoelastic properties in this study is the covalently crosslinked protein network. The composition of atherosclerotic plaque varies of course with its histological classification and according to its developmental stage, with earlier plaque stages exhibiting in general a higher fatty content and a softer structure [39]. As the plaque matures, calcification progresses, effectively hardening the plaque formation. The recipe of the artificial plaque can of course be adapted to the developmental stage of interest. Our focus was based on later-stage plaques, which is why we settled for a lower lipid mix content and higher calcification, but these levels can be adapted for the experimental question at hand. As the above-mentioned results illustrate, crucial for obtaining tough elastic plaque structures is the use of higher gelatin contents and the IDC crosslinking procedure.

### 3.2. Effect of Incubation on Plaque Stability

After showing that the required plaque dimensions could be obtained with the casting procedure, it was necessary to characterize the behavior of the plaque model upon prolonged incubation at a physiological temperature, thus indicating if the models are suitable for potential in vivo application. Long-term incubation in sterile phosphate-buffered saline at 37 °C led to pronounced swelling of the plaque produced by the DC procedure, which is shown by the weight increase in the DC plaques in Figure 5D, followed by a sudden dissolution of the hydrogel structure after day 14. The DC model is not suitable for modeling atherosclerotic plaques, as it does not display sufficient mechanical properties, in addition to exhibiting poor shape fidelity and stability upon longer incubation. Crosslinking by the DDC and IDC procedures resulted in plaques that showed a slight decrease in weight after the first three days of incubation, which was caused by the dissolution of non-crosslinked gelatin material. After 3 days, the plaques’ mass was stable for the duration of the experiment (49 days). The shrinking of the plaque model was also characterized microscopically as shown in Figure 5A,B. In general, the plaques produced by the DDC and IDC methods displayed shrinking, characterized by a slight decrease in length and outer and inner diameters (Figure 5C). Especially relevant is the decrease in the inner diameter, which was -7% for the DDC model and -9% for the IDC crosslinking. This behavior is beneficial for the intended application of modeling stenting complications, as it will lead to a tighter contact between the stent and plaque.

### 3.3. Cytotoxicity

For the AAP to be used in future experiments for chronic studies in animals, testing for cytotoxicity was very important to the authors. The AAP does not replicate the high inflammatory environment present in the setting of this disease. Even though the plaque has no cellular components, the biological aspect of the material with all its interactions does.

The extraction medium did not directly reduce MTT, as verified in the cell-free samples. Morphologically, the cells showed no signs of cytotoxicity after treatment with 100% of the plaque extract (Figure 6A) compared to the viability control (Figure 6B), whereas in the dead control treated with 0.01% *v*/*v* Triton X-100 (Figure 6C), a clear rounding of the cells was seen. Although a concentration-dependent reduction in viability was observed in the cells treated with different percentages of the extraction medium compared to the viability control, the viability of all samples was above 70% and was therefore classified as non-cytotoxic according to ISO 10993-5 (Figure 7).

### 3.4. Implantability in Porcine Coronary Artery

For the implantation of the AAP into the LAD, we took porcine hearts (aged between 6 and 9 months) from a local slaughterhouse. Five hearts received a plaque implantation.

The AAP was pushed onto the PTCA balloon catheter (CoCr Genous 2.75 × 18) including a bare metal stent (Amazonia CroCo Stent 6–20 mm) and fixed with 4 bars of pressure. The implantation system (PTCA balloon + stent + plaque) was via the ostium through a guide catheter and along the previously set guide wire to the LAD. The implantation system was placed at the target area (medial area of the LAD) and expanded at 20 bars for 20 s. After checking the correct position, the stent implantation system including catheter and guide wire was removed. Figure 8 shows the application of the AAP in the LAD in the imaging of a robot-assisted angiography system (Artis pheno^®^, Siemens Healthcare, Forchheim, Germany). 

## 4. Discussion

In this study, we created an AAP as a device to induce occlusions in coronary arteries that can correspond with the surrounding tissue and can be applied in the blood vessel of interest. This method provides a short-term model to simulate a partial stenosis with reduced perfusion and helps to test different treatment methods curing CHD in a certain vessel. An improved understanding of the mechanical properties of diseased coronary tissue may enhance our understanding of the pathologic influence of the stenosis and treatment on the rest of the vessel.

Atherosclerotic plaques vary in their composition and surrounding features like calcification, neovascularization, intraplaque hemorrhage, and thrombosis [48]. Depending on the focus of the research, it is important to modulate these necessary features. A clinical trial for the chronical changes after AAP application is necessary. Also, rupture and erosion of such plaques are key mechanisms responsible for the development of cardiovascular events, which are related to the stress levels within the plaque cap and need further investigations.

The material properties represent a simplified elastic material, not a strain-stiffening material. Also, the AAP geometry is a cylinder, not a stenosis commonly found in human atherosclerotic vessels. For better handling during implantation and the associated reproducible results, we developed the AAP as a concentric body, whereas natural atherosclerotic plaque material occurs in a wide variety of forms, mainly eccentrically. This shift of the center of the circle or the fulcrum can, of course, also have considerable consequences for the expansion of the plaque and the penetration force on the surrounding tissue.

Atherosclerotic tissues are complex inhomogeneous tissues with spatial gradients in material properties across components. While human plaque tissues were tested, only the storage modulus is reported, which can be limited. A more complete characterization is expected for both human tissues and artificial plaques for standardized testing and characterization protocols [7].

It is necessary to test whether the material composition of the AAP can be optimized for the research of atherosclerosis. Especially regarding the introduction of the plaques in an in vivo model via catheters/sheaths in the carotid or femoral artery, a stable material is required that is not destroyed during the application process and comes to rest well in the LAD.

Given that atherosclerotic tissue is a build-up of cells, necrotic debris, and more within the arterial wall and does not rest on the arterial surface, our ex vivo experiment does not fully represent the in vivo environment. The interventional and surgical procedure was used to test the handling for the APP implantation through guided wires into the LAD. This information is very relevant for clinical handling and in vivo procedures and could provide more data to demonstrate the efficacy of the AAP in situ.

This study compares the AAP stiffnesses with different crosslinking procedures to plaques obtained from three patients. This comparison is very important, but it would be helpful to compare the AAP formulations to patient plaques of known histological composition. 

## 5. Conclusions

Artificial atherosclerotic plaques with clinical similarities to human atherosclerotic plaques were produced according to their physical and chemical properties. The recipe of the gelatin matrix from porcine skin, mixed with the lipid component (cholesterol and phospholipids from soybean) and HA and finely grained CaCO_3_ are most promising to copy these plaques and create a helpful device for future atherosclerotic infarction modeling. Although the surrounding coronary tissue cannot be implicated, it is suspected that changes can also be observed in chronic follow-ups. The building of artificial plaque models is one part of the work; the application of these plaques in living organisms is another. The implantation of this artificial plaque in a living organism should be the next mandatory step for the establishment of an arteriosclerosis model.

We are convinced that an application of the AAP into a blood vessel provides a great technique for future medical trials and could help to test whether the pressure exerted intraluminally of a *Percutaneous coronary intervention* or the creation of a bypass anastomosis (connection of two vessels) leads to a reduced blood flow in the distal *vasa vasorum* (the blood vessels supplying the vascular wall), a disruption of the lymphatic drainage, or damage to the innervation.

## Figures and Tables

**Figure 1 biomimetics-09-00197-f001:**
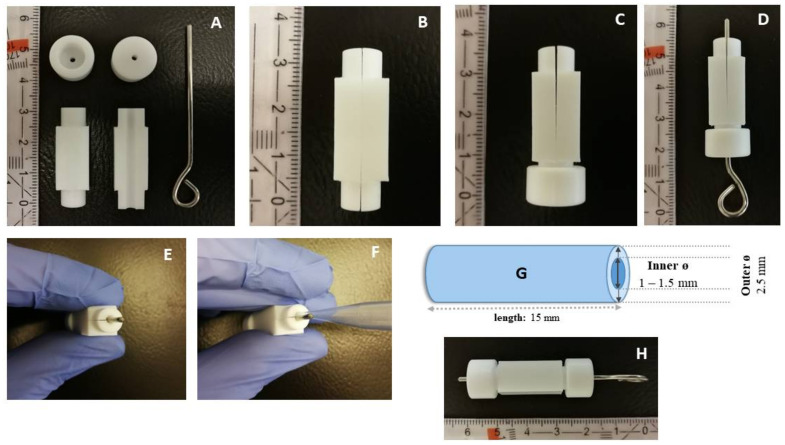
The casting mold consists of two hollow half-cylinders, two caps, and a metal rod (**A**). The two hollow half-cylinders are held together by one cap (**B**,**C**), after which the rod is inserted, defining the hollow space (**D**,**E**). The artificial plaque mixture is pipetted into the mold (**F**), leading to the formation of a hollow cylindrical plate with certain dimensions, which could subsequently be cut to the desired diameters (**G**). The assembled casting mold is shown in (**H**).

**Figure 2 biomimetics-09-00197-f002:**
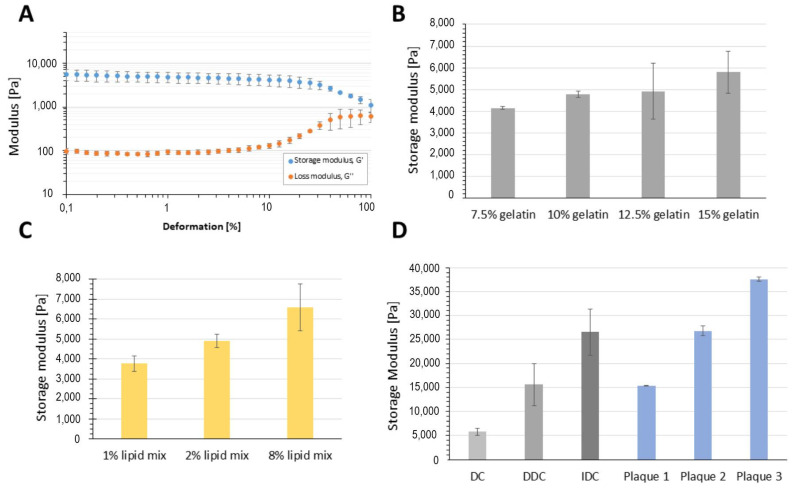
Effect of recipe composition and crosslinking procedure on final artificial plaque properties. (**A**) Viscoelastic mechanical properties of the model plaques were analyzed by amplitude sweeps. (**B**) The increase in gelatin concentration was studied while keeping all other components at the same level (2% lipid mix, 5% HA, and crosslinking procedure DC). (**C**) The influence of lipid mix variation in the plaque recipe (10% gelatin, 5% HA, and DC). (**D**) The largest impact on plaque mechanical properties results from the process of crosslinking, with the indirect crosslinking procedure (IDC) achieving shear modulus properties similar to donor plaques; all recipes here were 15% gelatin, 2% lipid mix, and 10% HA. The values of the shear modulus (frequency: 0.5% amplitude and 1 Hz) were taken from the linear viscoelastic region (LVE) of the amplitude sweep; all values were at least triplicates at 37 °C.

**Figure 3 biomimetics-09-00197-f003:**
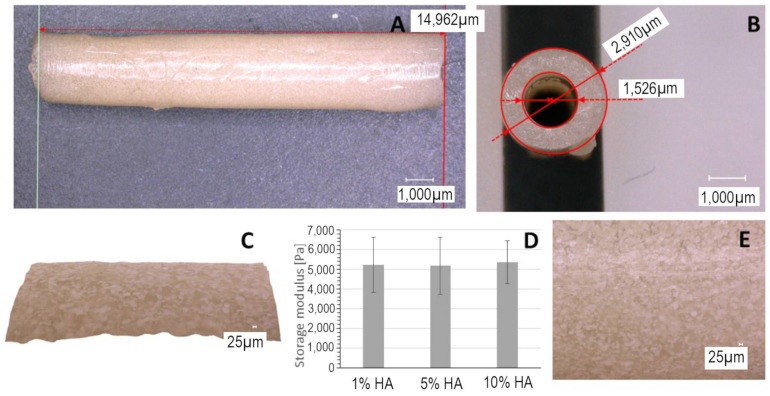
Microscopic examination of the manufactured plaque (12.5% gelatin, 2% lipid mix, and 10% HA), magnification 200×. (**A**) and (**B**) the desired proportions were replicated by the casting method. (**C**) Topologic examination. (**D**) The variation in HA amount does not change mechanical properties of the plaque. (**E**) The insoluble HA finely dispersed into the hydrogel network.

**Figure 4 biomimetics-09-00197-f004:**
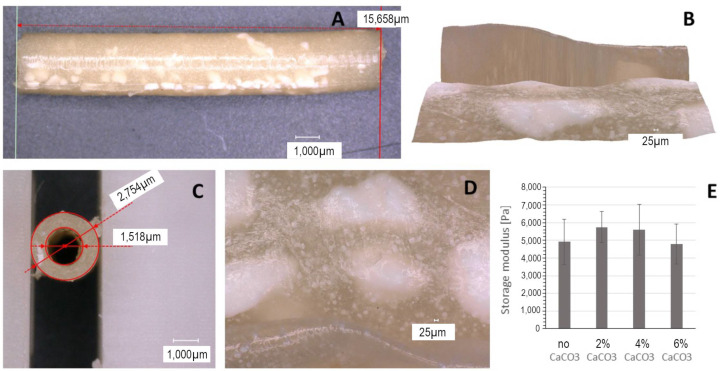
Microscopic examination of the manufactured plaque (12.5% gelatin, 2% lipid mix, 5% HA, and 4% CaCO_3_) magnification (200×). (**A**) The plaque visualized along its length. (**B**) Topologic examination showing irregular structure caused by carbonate islands. (**C**) Diameter measurements. (**D**) Carbonate grains are coarser and larger than HA. (**E**) Variation in calcium carbonate content does not impact plaque mechanical properties significantly.

**Figure 5 biomimetics-09-00197-f005:**
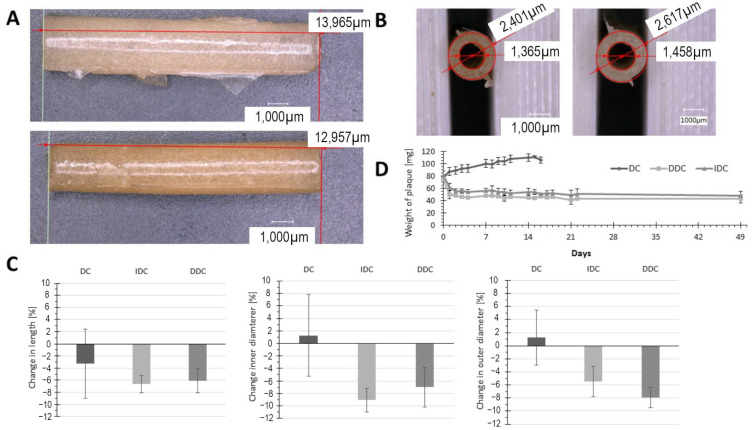
Long-term stability of the artificial plaque model. (**A**) Microscopic examination of the length of the manufactured plaque with the DDC procedure directly after casting (top) and after 3 days incubation at 37 °C (bottom). (**B**) Inner and outer diameters after casting (left) and after 3 days incubation at 37 °C (right). (**C**) Quantification of plaque model dimension changes for each crosslinking procedure. (**D**) Weight changes in plaque models from each crosslinking procedure over long-term incubation at 37 °C in sterile PBS.

**Figure 6 biomimetics-09-00197-f006:**
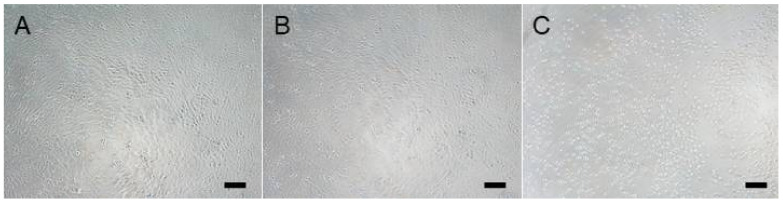
V79MZ cells 24 ± 2 h after treatment with (**A**) 100% plaque extraction medium, (**B**) fresh medium without extraction medium as viability control, and (**C**) fresh medium with 0.01% *v*/*v* Triton X-100 as dead control. Scale bars = 100 µm.

**Figure 7 biomimetics-09-00197-f007:**
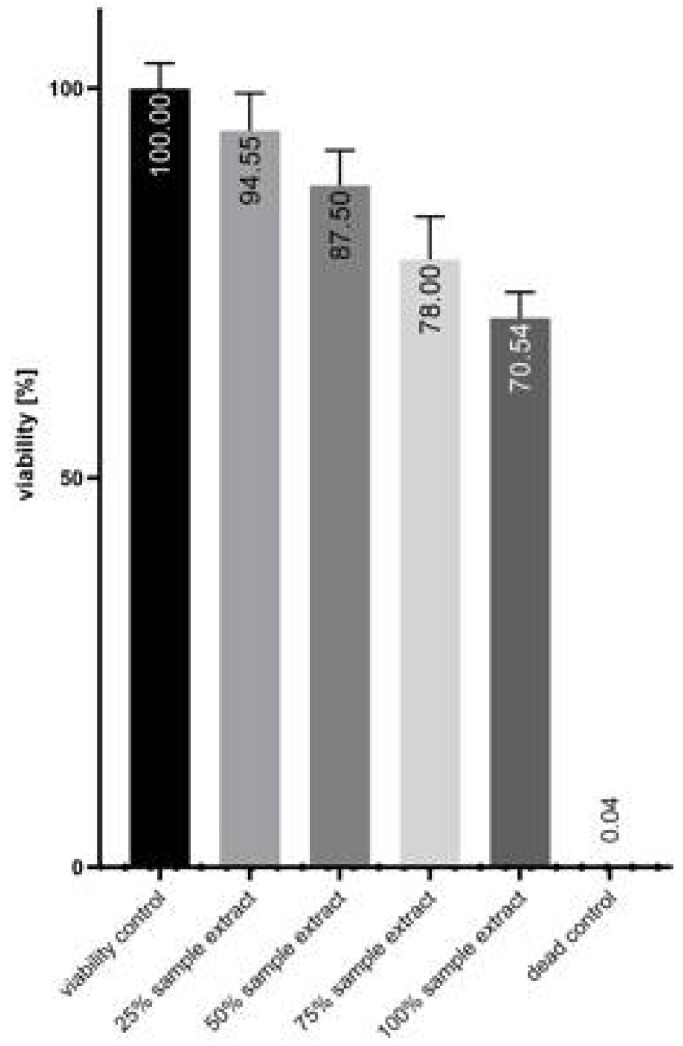
V79MZ viability after treatment with rising percentages of plaque extraction medium and corresponding controls. Shown are mean values ± standard deviation of 4 technical replicates, n = 1.

**Figure 8 biomimetics-09-00197-f008:**
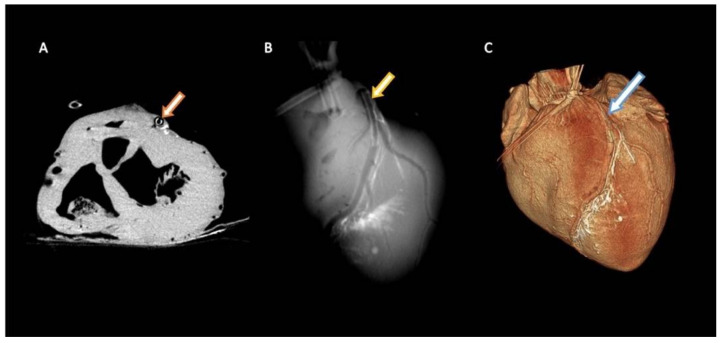
Axial view (**A**), maximum intensity projection (MIP) (**B**), and volume-rendering technique (VRT) (**C**) of the inflated porcine heart at the level of the previously inserted, artificially created atherosclerotic plaque into the left anterior descending artery (LAD). Note the contrasted cardiac lymphatic vessels adjacent to the LAD.

## Data Availability

The basic data used to support the results of this study were not made available due the interdisciplinary collaboration of several institutions. All research institutions involved continue to work on improving the artificial atherosclerotic plaque model. All relevant raw data are provided in this manuscript.

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
