# Peer review of "A Novel Artificial Coronary Plaque to Model Coronary Heart Disease"

_biomimetics, 2024, doi:10.3390/biomimetics9040197_

Round 1

Reviewer 1 Report

Comments and Suggestions for Authors

The production of functional elements for use in medicine to replace various parts of the human body has been, is and will be an urgent task for humanity. Over time, materials, approaches to the manufacture of implants and methods of their implantation change. The manufacture of functional elements from polymer and other materials requires very careful characterization, modeling of manufacturing and further testing of the finished product. Rheology in this case is one of the methods without which it is difficult to assess the characteristics of the material and select optimal molding conditions. In this paper, the authors propose the use of gelatin matrix from porcine skin mixed with the lipid component of cholesterol and phospholipids from soybean, hydroxyapatite and finely grained calcium carbonate to create an artificial plaque.

The authors describe in detail the experimental part of the work, and here questions arose regarding the correctness of the dimensions in the diagram in Fig. 1. and stability period (49 days)?

It is advisable to avoid such phrases in the manuscript "We designed, produced, and tested different materials."

"Conclusion: This artificial atherosclerotic plaque model works as a simulating
tool for future medical testing and could be crucial for further specified research on coronary artery
disease and is going to help to provide information about the optimal interventional and surgical
care of the disease." - Conclusions should be at the end of the manuscript. I propose to delete this part.

Figure 1G. Need to check the cylinder length, 15 mm?

Figure 2. At what frequency were the storage modulus values determined?

The donor plaques were also analyzed by oscillatory rheology and displayed highly variable viscoelastic moduli as can be seen in Fig 1C. you need to check the picture number.

Figure 3. Correct the caption (“the variation in HA amount does not change mechanical properties of the plaque”). Add scale bars to the picture and increase the font size. Figure 4. same thing.

Figure 5. The quality of the figure is very low.

The conclusions reflect the results described in the work and here I would recommend that the authors replace this part with “We are convinced...”

Author Response

X The authors describe in detail the experimental part of the work, and here questions arose regarding the correctness of the dimensions in the diagram in Fig. 1. and stability period (49 days)?
The cylinder here is the desired size/length of the plaque we wanted in the end and not the mold shown in the other pictures. The mold was slightly longer because the plaque could be cut individually after modeling. (the information was also added again in the caption).

The stability period was 49 days.

X It is advisable to avoid such phrases in the manuscript "We designed, produced, and tested different materials."
this part is deleted

X "Conclusion: This artificial atherosclerotic plaque model works as a simulating tool for future medical testing and could be crucial for further specified research on coronary artery disease and is going to help to provide information about the optimal interventional and surgical
care of the disease." - Conclusions should be at the end of the manuscript. I propose to delete this part.

The conclusion is part of the abstract and is therefore arranged in chronological order at the beginning of the manuscript.

X Figure 1G. Need to check the cylinder length, 15 mm?

The cylinder here is the desired size/length of the plaque we wanted in the end (“plaque of specified dimensions”) and not the mold shown in the other pictures. The mold was slightly longer because the plaque could be cut individually after modeling. (the information was also added again in the caption:

Figure 1: The casting mold consists of two hollow half-cylinders, two caps and a metal rod (A). The two hollow half-cylinders are held together by one cap (B), (C), after which the rod is inserted defining the hollow space (D), (E). The artificial plaque mixture is pipetted into the mold (F), leading to the formation of a hollow cylindrical plate with certain dimensions, which could subsequently be cut to the desired diameters (G). The assembled casting mold is shown in (H).

X Figure 2. At what frequency were the storage modulus values determined?

The frequency used in all my rheometer experiments was 1 Hz and was not changed. This is already in chapter 2.4. (Information added to Fig.2)

“frequency (0.5% amplitude and 1 Hz)”

X The donor plaques were also analyzed by oscillatory rheology and displayed highly variable viscoelastic moduli as can be seen in Fig 1C. you need to check the picture number.

Figure 1C shows the casting mold. - The link in the text for the donor plaques was corrected (Fig. 2D)

X Figure 3. Correct the caption (“the variation in HA amount does not change mechanical properties of the plaque”). Add scale bars to the picture and increase the font size. Figure 4. same thing.

As requested we did adjustments on the Figures 3 and 4 (font size, scale bars)

X Figure 5. The quality of the figure is very low.

The Figure was upgraded with a higher resolution

X The conclusions reflect the results described in the work and here I would recommend that the authors replace this part with “We are convinced...”

In the conclusion you already find the part “We are convinced…” - Please explain or add to your comment.

Reviewer 2 Report

Comments and Suggestions for Authors

Dear authors, the manuscript is of interest to scientists in the field of disease modeling, cardiac disease and material science. The introduction is clearly written. I would like to suggest some points for further improvevment:

- The cell toxicity test is very useful, however I would suggest to include a high magnification into figure 6. In addition, I would recommend to include a better description in the figure legend of figure 7, for example statistics, sample size.

- A blood flow measurement after in vivo implantation would increase the impact of the study.

- Please check the manuscript for spelling and grammar errors, some sentences are written twice, for example line 399.

Comments on the Quality of English Language

Some minor errors.

Author Response

- The cell toxicity test is very useful, however I would suggest to include a high magnification into figure 6. In addition, I would recommend to include a better description in the figure legend of figure 7, for example statistics, sample size.

Thank you for your valuable comment. Unfortunately, it is not possible to provide a higher magnification in Figure 6, as we did not capture images at that level of detail. However, we have updated the figure legend for Figure 7 as requested. (Please see the corrected file in the attachment)

- A blood flow measurement after in vivo implantation would increase the impact of the study.

For this project we tested the ex vivo process only, like modeling, handling and implantability of the artificial plaque. We also think the future in vivo testing is necessary and very important for further adjustments. 

- Please check the manuscript for spelling and grammar errors, some sentences are written twice, for example line 399.

The duplicate sentence was subsequently removed
